# Globus pallidus dynamics reveal covert strategies for behavioral inhibition

Bon-Mi Gu[1], Robert Schmidt[2], Joshua D Berke[1,3]*

[1]Department of Neurology, University of California, San Francisco, San Francisco, United States; [2]Department of Psychology, University of Sheffield, Sheffield, United Kingdom; [3]Department of Psychiatry; Neuroscience Graduate Program; Kavli Institute for Fundamental Neuroscience; Weill Institute for Neurosciences, University of California, San Francisco, San Francisco, United States

**Abstract** Flexible behavior requires restraint of actions that are no longer appropriate. This behavioral inhibition critically relies on frontal cortex - basal ganglia circuits. Within the basal ganglia, the globus pallidus pars externa (GPe) has been hypothesized to mediate selective proactive inhibition: being prepared to stop a specific action, if needed. Here we investigate population dynamics of rat GPe neurons during preparation-to-stop, stopping, and going. Rats selectively engaged proactive inhibition towards specific actions, as shown by slowed reaction times (RTs). Under proactive inhibition, GPe population activity occupied state-space locations farther from the trajectory followed during normal movement initiation. Furthermore, the state-space locations were predictive of distinct types of errors: failures-to-stop, failures-to-go, and incorrect choices. Slowed RTs on correct proactive trials reflected starting bias towards the alternative action, which was overcome before progressing towards action initiation. Our results demonstrate that rats can exert cognitive control via strategic adjustments to their GPe network state.

*For correspondence:
joshua.berke@ucsf.edu

Competing interests: The authors declare that no competing interests exist.

## Introduction

Our capacity for self-restraint is critical for adaptive behavior. Dysfunctions in behavioral inhibition are involved in many human disorders, including drug addiction (*Ersche et al., 2012*). A standard test of behavioral inhibition is the stop-signal task (*Logan et al., 1984*; *Verbruggen et al., 2019*), in which subjects attempt to respond rapidly to a Go cue, but withhold responding if the Go cue is quickly followed by a Stop cue. The stop-signal task has been invaluable for revealing specific cortical-basal ganglia mechanisms involved in both movement initiation ('Going'; e.g. *Hanes and Schall, 1996*) and inhibition ('Stopping'; e.g. *Aron and Poldrack, 2006*; *Eagle et al., 2008*). 'Reactive' inhibition – making quick use of a Stop cue – appears to involve at least two distinct mechanisms (*Schmidt and Berke, 2017*): a rapid Pause process mediated via the subthalamic nucleus (STN; *Aron and Poldrack, 2006*; *Schmidt et al., 2013*) followed by a Cancel process achieved through pallidostriatal inhibition (*Mallet et al., 2016*).

Behavioral inhibition can also be 'proactive': restraint of actions, in advance of any Stop cue. Proactive inhibition may be particularly relevant to human life (*Aron, 2011*; *Jahanshahi et al., 2015*). Whereas reactive inhibition typically involves a global, transient arrest of actions and thoughts (*Wessel and Aron, 2017*), proactive inhibition can be selectively directed to a particular action (*Cai et al., 2011*). A key behavioral signature of proactive inhibition is slowing of reaction times (RTs) for that action, when the anticipated Stop cue does not actually occur (e.g. *Verbruggen and Logan, 2009*; *Chikazoe et al., 2009*; *Zandbelt et al., 2013*). This overt behavioral signature presumably relies on covert shifts in information processing, yet the nature of these shifts is unclear. In some studies, fitting of models to behavioral data has suggested that slowed RTs reflect raising of a

decision 'threshold' (*Verbruggen and Logan, 2009*; *Jahfari et al., 2012*), but other studies have found evidence for a slower rate of progression toward threshold instead (*Dunovan et al., 2015*).

The neural circuit mechanisms by which proactive control is achieved are also not well understood. It has been proposed that proactive inhibition critically depends on the basal ganglia 'indirect' pathway via GPe (*Aron, 2011*; *Jahanshahi et al., 2015*; *Dunovan et al., 2015*). Yet clear support for this hypothesis is sparse (*Majid et al., 2013*). There have been few electrophysiological studies of proactive inhibition at the level of individual neurons (*Chen et al., 2010*; *Pouget et al., 2011*; *Hardung et al., 2017*; *Yoshida et al., 2018*), and to our knowledge none in GPe. We therefore targeted GPe (often called simply GP in rodents) for investigating neural mechanisms of proactive control.

We also wished to integrate a dynamical systems approach into the study of behavioral inhibition, and the basal ganglia. Analysis of the collective dynamics of motor cortex neurons has provided insights into various aspects of movement control, including how brain networks may prepare actions without prematurely triggering them (*Kaufman et al., 2014*), and the origins of RT variability (*Afshar et al., 2011*). We demonstrate below that the analysis of GPe population activity can reveal distinct covert strategies underlying overt manifestations of proactive control.

## Results

### Action initiation is slower when a stop cue is expected

We trained rats in a modified version of our stop-signal task (*Figure 1A*; *Leventhal et al., 2012*; *Schmidt et al., 2013*; *Mallet et al., 2016*). Freely-moving rats poked their noses into a hole and maintained that position for a variable delay (500–1250 ms) before presentation of one of two Go cues (1 kHz or 4 kHz tone), instructing leftward or rightward movements respectively into an adjacent hole. If initiated rapidly (RT limit <800 ms), correct movements triggered delivery of a sugar pellet reward from a separate food hopper. On some trials the Go cue was quickly followed by a Stop cue (white noise burst), indicating that the rat instead needed to maintain its nose in the starting hole (for a total of 800 ms after Go cue onset) to trigger reward delivery. The delay between Go and Stop cue onsets (100–250 ms) ensured that stopping was sometimes successful and sometimes not. As expected, Failed Stop (error) trials had similar RTs to the faster part of the Go trial RT distribution (*Figure 1B*). This is consistent with the basic 'race' conceptual model of reactive inhibition (*Logan et al., 1984*): failures-to-stop typically occur when an underlying Go process evolves more quickly than average (*Schmidt et al., 2013*), and thus wins the race against a separate Stop process.

To probe selective proactive inhibition we used a 'Maybe-Stop versus No-Stop' approach (*Aron and Verbruggen, 2008*). The three possible starting holes were associated with different Stop cue probabilities (*Figure 1C*): no possibility of Stop cue; 50% probability that a left Go cue (only) will be followed by the Stop cue; or 50% probability that a right Go cue (only) will be followed by the Stop cue. Our index of proactive inhibition was a preferential increase in RT for the Maybe-Stop direction, compared to the No-Stop conditions. Among rats that began learning this task variant, approximately half acquired clear proactive inhibition within 3 months of training (see Materials and methods), and were thus considered eligible for electrode implantation. Here we report behavioral and neural results for six rats for which we were able to obtain high-quality GP recordings as rats engaged proactive control.

We selected for further analysis those behavioral sessions (n = 63) with a significant proactive inhibition effect (i.e. longer RT when a Stop cue might occur; one-tail Wilcoxon rank sum test, p<0.05) and distinct GP single units (n = 376 neurons included). Prior work has shown particular basal ganglia involvement in the control of contraversive orienting-type movements (i.e. directed towards the opposite side; *Carli et al., 1985*; *Isoda and Hikosaka, 2008*; *Schmidt et al., 2013*; *Leventhal et al., 2014*). We therefore focused on proactive control of movements contraversive ('contra') to the recorded cell locations; for example we included a left GPe cell only if the rat demonstrated proactive control for rightward movements during that recording session. For included sessions, median RT for correct contra movements was 251 ms when the Stop cue could not occur (No-Stop), and 385 ms when the Stop cue could occur (Maybe-Stop) but did not. Results from all sessions, and from individual animals, are shown in *Figure 1—figure supplement 1*.

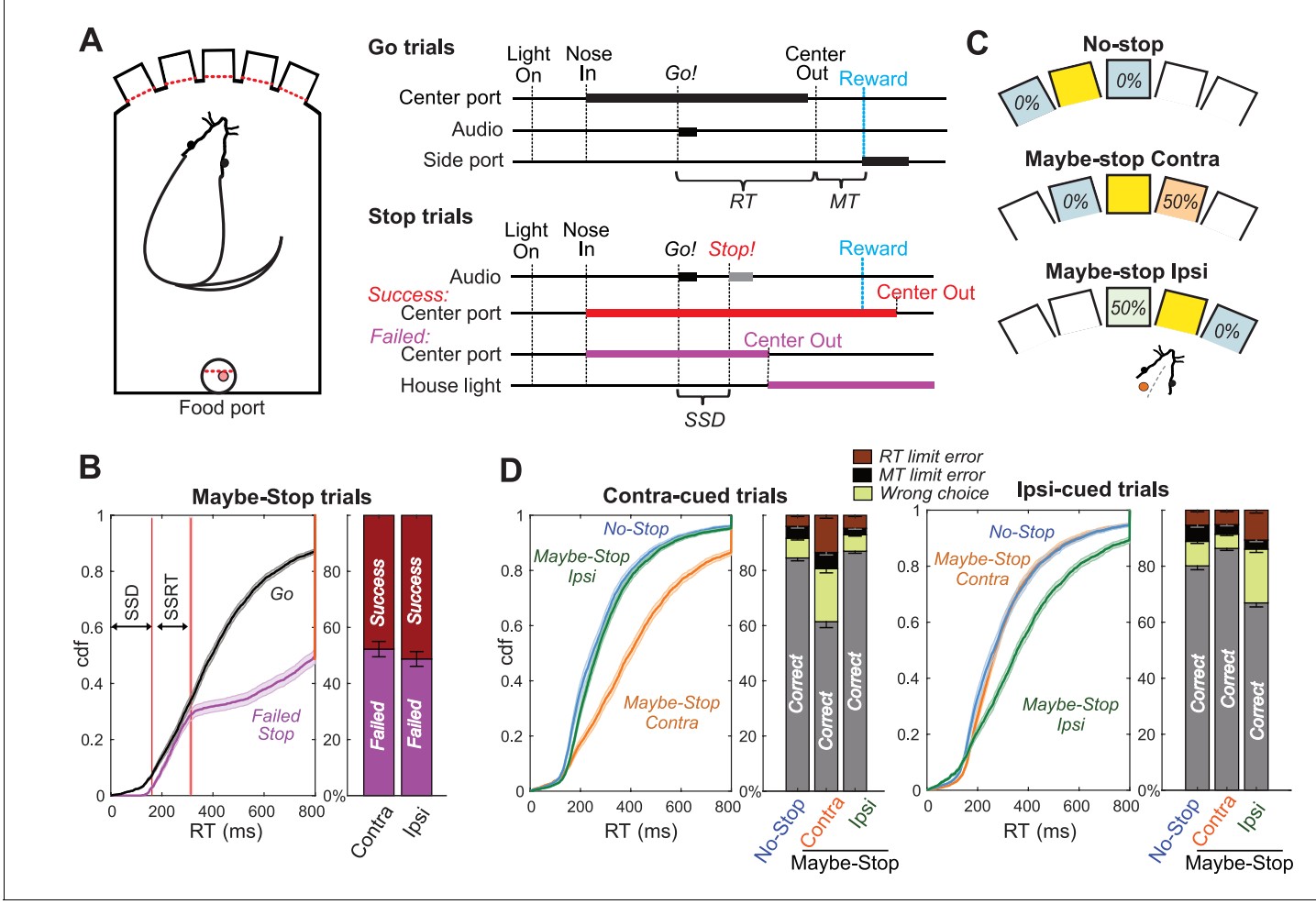

**Figure 1.** Reactive and Proactive Behavioral Inhibition. (**A**) Left, operant box configuration; right, event sequence for Go and Stop trials. RT, reaction time; MT, movement time; SSD, stop-signal delay; Reward, delivery of a sugar pellet to the food port. (**B**) Left, distributions of Go and Failed-Stop RTs (on Maybe-Stop trials; shading, S.E.M. across n = 63 sessions). Failed-Stop RTs are similar to the faster part of the Go RT distribution, consistent with the 'race' model in which a relatively-fast Go process produces failures to stop. The tail of the Failed-Stop distribution (RT >500 ms) is presumed to reflect trials for which rats successfully responded to the Stop cue, but then failed to maintain holding until reward delivery (see *Leventhal et al., 2012*; *Schmidt et al., 2013*; *Mayse et al., 2014*). Right, proportions of failed and successful Stop trials after Contra and Ipsi Go cues. Error bars, S.E.M. across n = 63 sessions. (**C**) Trial start location indicates stop probabilities (locations counterbalanced across rats). In this example configuration recording from left GP, starting from the middle hole indicates the Maybe-stop Contra condition: Go cues instructing rightward movements might be followed by a Stop cue, but Go cues instructing leftward movements will not. (**D**) Proactive inhibition causes selective RT slowing for the Maybe-Stop direction (two-tail Wilcoxon signed rank tests on median RT for each session: contra cues in Maybe-Stop-contra versus No-Stop, z = 7.7, p=1.15 × 10$^{-14}$; ipsi cues in Maybe-Stop-contra versus No-Stop, p=0.32). Additionally, under selective proactive inhibition rats were more likely to fail to respond quickly enough (RT limit errors; Wilcoxon signed rank tests, z = 7.2, p=5.41 × 10$^{-13}$) and to select the wrong choice (uncued action direction; Wilcoxon signed rank tests, z = 7.0, p=2.59 × 10$^{-12}$). Error bars, S.E.M. across n = 63 sessions. Only trials without a Stop cue are included here. *RT limit error* = Nose remained in Center port for >800 ms after Go cue onset; *MT limit error* = movement time between Center Out and Side port entry >500 ms.

The online version of this article includes the following figure supplement(s) for figure 1:

**Figure supplement 1.** Behavioral data for all sessions and for each individual animal.

RT slowing due to proactive inhibition was highly selective to the Maybe-Stop direction (*Figure 1D*; *Figure 1—figure supplement 1*; for Maybe-Stop-Contra trials without a Stop cue, median ipsiversive ('ipsi') RT was unslowed at 264 ms). The Maybe-Stop condition was also associated with an increase in errors (*Figure 1D*), in particular not responding quickly enough to the Go cue that might be followed by Stop (RT limit error; RT >800 ms) and making the wrong choice (incorrect action selection). These error types are examined further below.

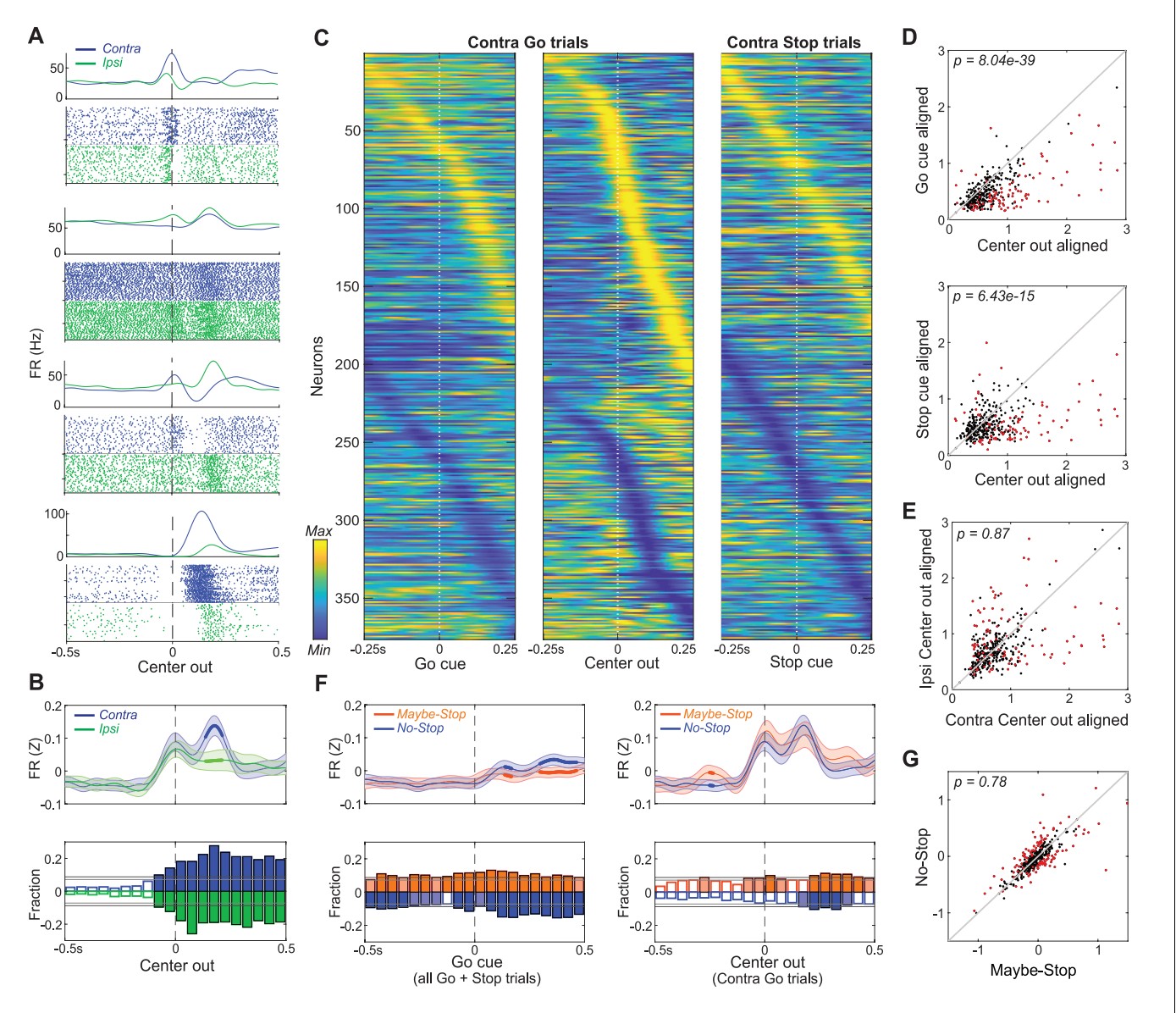

**Figure 2.** Movement-related activity of individual GP neurons. (**A**) Four examples of single neurons, showing average firing rates (top) and spike rasters (bottom) aligned on movement onset (Center Out; correct No-Stop trials only). Activity for contra-, ipsi- movements are shown in blue and green respectively. (**B**) Top, averaged, Z-scored firing of GP cells around Center Out; time points when activity distinguishes movement direction are shown with thicker lines. Shaded band, +- S.E.M across n = 376 neurons. Bottom, fraction of neurons whose firing rate significantly distinguishes movement direction, across time (t-test for each neuron in each 50 ms bin, p<0.05). Higher firing rate for contra-, ipsi- shown in blue, green respectively. Horizontal grey lines indicate thresholds for a significant proportion of neurons (binomial test, p<0.05 without or with multiple-comparisons correction respectively) and bins that exceed these thresholds are filled in color. Many GP cells encoded movement direction even before Center-Out; this is less obvious after averaging. (**C**) Firing pattern of all GP cells (n = 376) on correct contra trials. Activity is scaled between minimum and maximum firing rate across alignments to Go cue (left), Center Out (middle) and the Stop cue (right). In each column cell order (top-bottom) is sorted using the time of peak deflection from average firing, separately for cells that showed bigger increases (top) or decreases (bottom). (**D**) GP population activity is more related to movements than cues. Scatter plots show peak deflections in firing rate (Z-scored) for each GP cell, comparing Center Out aligned data to Go cue aligned (top) or Stop cue aligned (bottom). Data included are 500 ms around alignment time. Indicated p-values are from Wilcoxon signed rank tests over the GP population; individual GP cells that showed significant differences are indicated with red points (t test, p<0.05). (**E**) Scatter plot indicates no overall movement direction bias. Same format, same statistical tests as D, but comparing peak deflections in Center Out aligned firing rate for contra, ipsi movements. (**F**) Top, comparing average firing between Maybe-Stop and No-Stop conditions. On left, data are aligned on Go cue, including all Maybe-Stop-Contra trials (including both contra- and ipsi-instructing Go cues and Stop trials). On right, data are aligned on Center-Out (and does not include Stop cue trials). Bottom, proportion of neurons whose firing rate is significantly affected by proactive inhibition (same format as B; bins exceeding p<0.05 threshold without multiple comparisons correction are filled in light color, bins exceeding corrected threshold are filled in dark color).

*Figure 2 continued on next page*

*Figure 2 continued*

Although GP neurons significantly distinguished Maybe-Stop and No-Stop conditions at multiple time points before the Go cue, there was no single time point at which the proportion of individually-significant neurons became large. (G) Comparison of individual cell activity in Maybe-Stop and No-Stop conditions, during the 500 ms epoch immediately before the Go cue.

The online version of this article includes the following figure supplement(s) for figure 2:

**Figure supplement 1.** Further details of GP recordings.

## GP firing rate changes related to movement onset and proactive inhibition

We recorded individual neurons (n = 376) from a wide range of GP locations (*Figure 2—figure supplement 1A*). As expected from prior studies (*DeLong, 1971*; *Brotchie et al., 1991*; *Gardiner and Kitai, 1992*; *Turner and Anderson, 1997*; *Arkadir et al., 2004*; *Gage et al., 2010*; *Shin and Sommer, 2010*; *Schmidt et al., 2013*; *Yoshida and Tanaka, 2016*; *Mallet et al., 2016*) GP neurons were tonically-active (mean session-wide firing rate, 28 Hz) with diverse, complex changes in firing patterns during task performance (*Figure 2A*). The majority of GP cells showed strongest firing rate changes (increases or decreases) when activity was aligned relative to movement onset, rather than to the Go or Stop cues (*Figure 2C,D*; see also *Figure 2—figure supplement 1B* for ipsi movement trials). Individual neurons showed greater changes for either contra or ipsi movements (*Figure 2A, B*), but these were about equally represented in the overall population (*Figure 2B,E*), and the average GP activity was similar for the two movement directions (at least until the movement was already underway; *Figure 2B*).

We next examined how the activity of individual GP neurons is affected by proactive inhibition. As rats waited for the (unpredictably-timed) Go cue, average firing was similar between Maybe-Stop and No-Stop conditions (*Figure 2F*), regardless of whether we examined cells that predominantly increase or decrease activity during movements (*Figure 2—figure supplement 1C*). We hypothesized that this average activity obscures a sizable GP subpopulation that consistently and persistently 'encodes' proactive control as rats wait. To search for this putative subpopulation we used a screening approach (similar to our prior work on reactive stopping; *Schmidt et al., 2013*; *Mallet et al., 2016*), comparing the Maybe-Stop-contra and No-Stop conditions. We did find that the fraction of GP cells that fired differently between these conditions was slightly greater than expected by chance (*Figure 2F*), consistent with GP involvement in proactive control. However, contrary to our hypothesis, we were not able to identify a clear subgroup of individual neurons that strongly and persistently distinguished between conditions (*Figure 2G*). Rather, proactive control was associated with altered activity in different subsets of GP neurons at various brief moments before the Go cue (*Figure 2— figure supplement 1D*).

## Population trajectories during movement selection and initiation

We next hypothesized that these GP firing rate differences, though subtle and diverse at the single-cell level, are coordinated to produce clear, interpretable changes in population dynamics. To observe these dynamics we began by reducing the dimensionality of population activity (*Cunningham and Yu, 2014*), using principal component analysis (PCA). For each neuron we included normalized, averaged firing rates for a 500 ms epoch around movement onset (separately for contra and ipsi movements; *Figure 3A*). We used the first 10 principal components (PCs; *Figure 3—figure supplement 1A*) to define a 10-dimensional state-space, with GP population activity represented as a single point in this space. For visualization we display the first 3 PCs (which together account for 71% of total population variance; *Figure 3B*), although statistical analyses used all 10 PCs.

Within state space, population activity was very similar for contra and ipsi movements at the Go cue (*Figure 3C*), and initially evolved in a common direction before progressively separating into distinct trajectories (*Video 1*). We used the common direction to define an '*Initiation Axis*', scaled between 0 (mean location at Go cue) and 1 (mean location at movement onset, Center Out). This

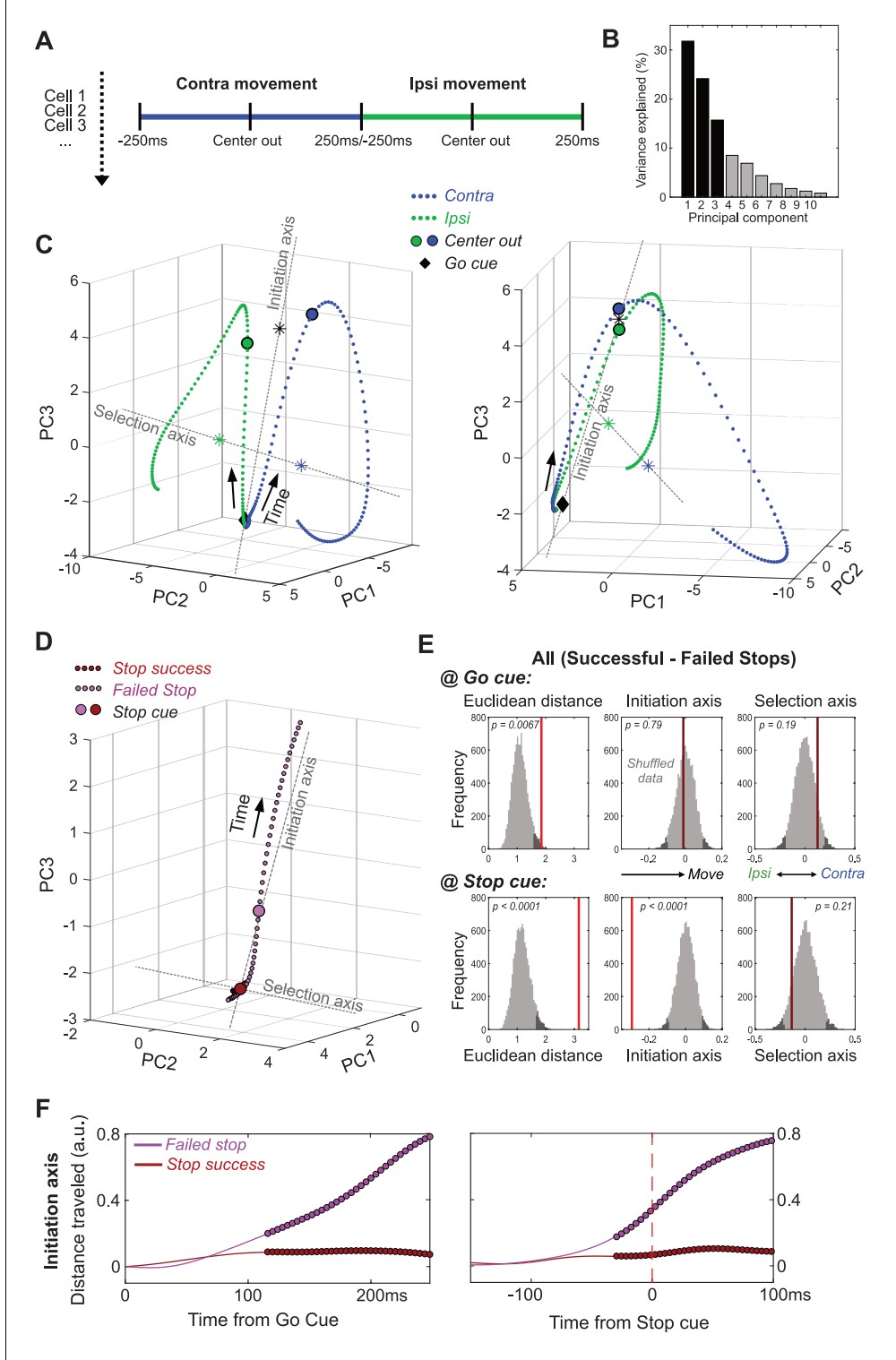

**Figure 3.** GP dynamics for Going and Stopping. (**A**) PCA was performed using averaged, normalized firing rates for each GP cell, in a 500 ms epoch around Center Out for contra and ipsi movements (concatenated). (**B**) Variance explained by each of the first 10 PCs. (**C**) GP state-space trajectories for contra and ipsi movements (blue, green) within the first 3 PCs, shown from two different angles. Each small dot along the trajectory is separated by 4 ms. Trajectories begin at a similar mean location at the Go cue (diamonds), and diverge gradually until Center Out (large circles) then rapidly thereafter. 'Initiation Axis' joins the average position at Go cue and the average position at Center Out (black asterisk). 'Selection Axis' joins the means of each trajectory, colored asterisks. (**D**) Comparing

*Figure 3 continued on next page*

*Figure 3 continued*

state-space trajectories for Successful- and Failed-Stop trials. Same format and PCA space as C, but plotting trajectories aligned on the Stop cue (including both contra and ipsi trials). Filled circles indicate epochs of significant Euclidean distance between two trajectories (permutation test on each 4 ms time bin, p<0.05). (**E**) Permutation tests of whether the state-space positions for Successful- and Failed-Stop trials are significantly different, at either the Go cue (top) or the Stop cue (bottom). Positions are compared either in the 10-D PC space (Euclidean distance) or along the Initiation or Selection Axes. Grey distributions show surrogate data from 10000 random shuffles of trial types. Dark grey, most extreme 5% of distributions (one-tailed for Euclidean, 2-tailed for others). Red vertical lines show observed results (bright red, significant; dark red, n.s.). (**F**) Distance travelled along Initiation Axis for successful and failed Stop trials, aligned on either Go cue (left) or Stop cue (right). Thicker lines indicate epochs of significant difference between successful and failed Stop trajectories (permutation test on each 4 ms time bin, p<0.05). On Failed stops (only), activity has already evolved substantially by the time of the Stop cue.

The online version of this article includes the following figure supplement(s) for figure 3:

**Figure supplement 1.** Principal Components.

---

allows us to quantify progression towards (or away from) movement onset. We used the difference between trajectories to define a '*Selection Axis*', scaled between −1 (mean of the ipsi trajectory) and +1 (mean of the contra trajectory). This allows us to quantify bias toward one movement direction or the other. Along both Initiation and Selection axes, change was not dominated by a small proportion of GP neurons. Instead, there were smaller contributions from many individual cells located throughout GP (*Figure 3—figure supplement 1AC-E*).

## Failed stops reflect earlier evolution of GP activity

We then considered how GP population activity is evolving when Stop cues occur. As noted above, standard race models of reactive stopping (*Logan et al., 1984*), together with prior data (*Schmidt et al., 2013*), suggest that failures-to-Stop occur when an underlying Go process evolves more quickly than average, and thus the Stop cue arrives too late. GP population activity was consistent with these ideas (*Figure 3D–F*). On successful-Stop trials GP activity showed little or no movement before the Stop cue. By contrast, on failed-Stop trials GP activity was in a significantly different state by the time of the Stop cue, having already evolved a substantial distance along the Initiation Axis (*Figure 3D*; includes both contra- and ipsi-cued trials). Thus, our observations of neural dynamics support hypothesized internal dynamics that determine whether we can react to new information, or are already committed to a course of action.

## When stop cues may occur, GP activity starts farther from movement initiation

Conceptually, the slowing of RT with proactive inhibition could reflect any of several distinct underlying changes (*Figure 4A*), that would manifest in GP dynamics in different ways. If slowing involves mechanisms 'downstream' of GP, we might observe no change in the GP population trajectory when aligned on the Go cue (*hypothesis 1*). Alternatively, the GP might be in a different state at the time the Go cue arrives. In particular, GP activity might start farther away from 'threshold' (in dynamical terms, farther from a subspace associated with movement initiation), and thus take longer to get there (*hypothesis 2*). Finally, proactive inhibition might cause GP activity to evolve differently *after* Go cue onset. Various, non-mutually-exclusive possibilities include a delayed start (*hypothesis 3*), slower

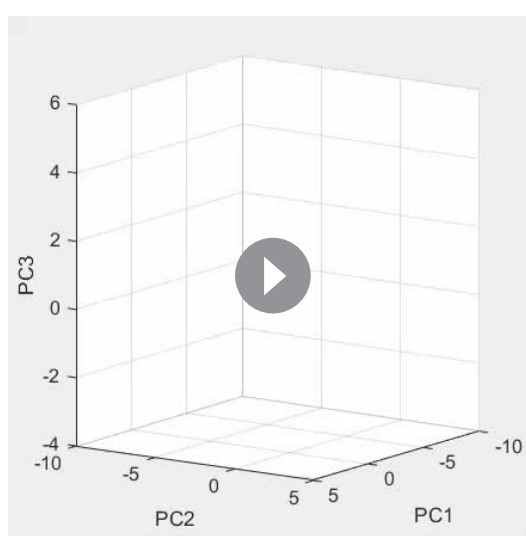

**Video 1.** Using movement-related trajectories through state-space to define Initiation, Selection Axes.
https://elifesciences.org/articles/57215#video1

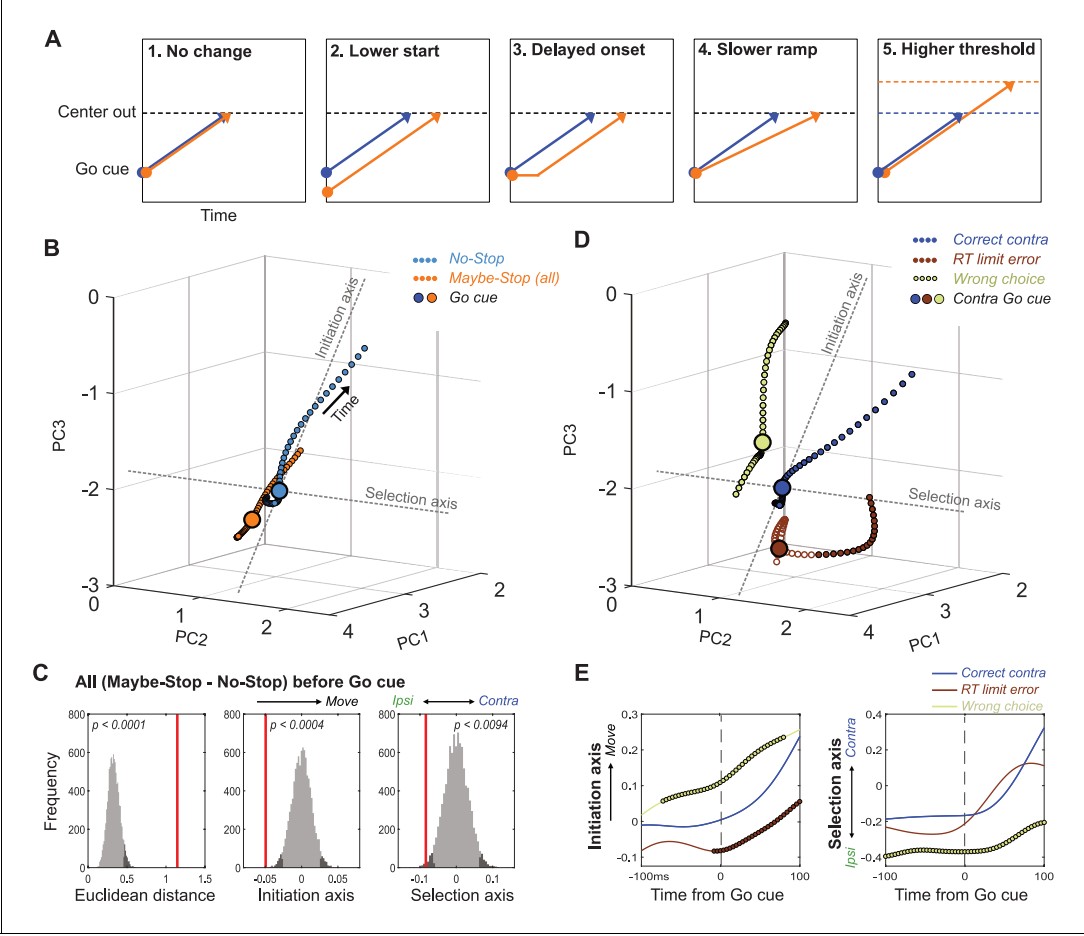

**Figure 4.** Distinct state-space positions at Go cue predict distinct outcomes. (**A**) Alternative concepts for proactive inhibition, illustrated using a simplified rise-to-threshold framework (*Brown and Heathcote, 2008*; *Noorani and Carpenter, 2016*; *Verbruggen and Logan, 2009*). (**B**). Comparison of GP population state between Maybe-Stop-Contra trials (including both contra- and ipsi-instructing Go cues and Stop trials) and No-Stop trials (± 100ms around Go cue; same state-space as *Figure 3*). Filled circles indicate epochs of significant Euclidean distance between two trajectories (permutation test on each 4 ms time bin, p<0.05). (**C**). Permutation tests (same format as *Figure 3*). Just before the Go cue (-100-0ms) the Maybe-Stop state was significantly shifted away from action initiation, and in the ipsi direction. (**D**) Breakdown of GP state for trials with contra Go cues, by distinct trial outcomes. (**E**) Quantification of D, comparing evolution of activity along Initiation and Selection Axes on correct contra trials (blue), incorrect action selections (light green) and RT limit errors (brown; failure to initiate movement within 800ms). Thicker lines indicate epochs of significant difference to the Correct trajectory (permutation test on each 4 ms time bin, p<0.05).

The online version of this article includes the following figure supplement(s) for figure 4:

**Figure supplement 1.** Neural population results for individual rats, and corresponding behavior.

**Figure supplement 2.** Trial-history dependence.

progress along the same trajectory (*hypothesis 4*), and/or a threshold that is shifted further away from the starting point (*hypothesis 5*). Of note, only hypothesis two predicts a change in the trajectory start location at the time of the Go cue (*Figure 4A*).

We compared GP population activity between Maybe-Stop and No-Stop conditions, immediately before the Go cue (−100 ms - 0 ms; including all trial subtypes). When proactive inhibition was engaged, GP activity occupied a significantly shifted location within state-space (*Figure 4B,C*). When examined along the Initiation axis (*Figure 4C*), the direction of this shift was consistent with a longer trajectory required for movements to begin (*hypothesis 2*). In other words, the brain can restrain actions by placing key circuits into a state from which actions are slower to initiate.

## Distinct state-space positions predict distinct types of errors

Proactive inhibition of contra movements also produced a significant shift along the Selection axis before the Go cue, in the direction associated with ipsi movements (*Figure 4C*). This suggests a preparatory bias against contra movements, when the contra-instructing Go cue may be followed by a Stop cue. To examine how starting position affects behavioral outcome, we examined how state-space location at the Go cue varies with distinct types of errors (*Figure 4D*). Failures to respond quickly enough to the Go cue (RT limit errors) were associated with starting farther away on the Initiation Axis (*Figure 4E*). By contrast, incorrect choices (ipsi movements despite contra cue) were associated with starting closer to movement initiation, together with a more-ipsiversive position on the Selection axis at Go cue (*Figure 4E*; *Video 2*). Thus, even while the animals are holding still, waiting for the Go cue, GP networks show distinctly-biased internal states that predict distinct subsequent behavioral outcomes.

## Overcoming a selection bias delays movement initiation

The starting ipsiversive bias on the Selection axis when contra actions might have to be cancelled can be overcome, as even on contra Maybe-Stop trials the rats usually made the correct choice. To examine how this occurs we compared neural trajectories for correct, contra Maybe-Stop and No-Stop trials (*Figure 5A*; only correct trials without Stop cues are included). Just before the Go cue on Maybe-Stop trials, rats showed no difference on the Initiation Axis but were significantly shifted on the Selection axis, in the ipsiversive direction (*Figure 5A,B*). After the Go Cue, movement on the Initiation axis was delayed compared to No-Stop trials, but movement on the Selection Axis occurred earlier (*Figure 5C*; *Video 3*). Thus, on correctly-performed Maybe-Stop trials the GP network engaged a dynamical sequence that was not observed on No-Stop trials: they first overcame a proactive bias towards the alternative action, before proceeding to initiate the action that had been cued.

Together our results indicate that, when faced with the challenging Maybe-Stop condition, rats adopt multiple, distinct, covert strategies. They can position neural activity farther from movement onset (on the Initiation Axis), but this produces limited hold violations – essentially making this a bet that the Stop cue will in fact occur. Alternatively, they can bias neural activity in the ipsi direction (on the Selection Axis). This delays contra choices, but also increases the rate of incorrect ipsi choices.

## Slower RTs can arise through multiple dynamic mechanisms

We considered the possibility that this apparent 'strategy' for proactive inhibition simply reflects the slower RT. In other words, is the distinct trajectory seen for correct Maybe-Stop trials also seen for slower No-Stop trials? Our data indicate that this is not the case. Comparing Maybe-Stop trials with No-Stop trials with the same RT (RT-matching) again showed different positions on the Selection Axis at Go cue (*Figure 5—figure supplement 1*). This difference was not seen when comparing slower and faster RTs within the No-Stop condition (*Figure 5D,E*). Rather, spontaneously-slower RTs appeared to arise through slower evolution along both Initiation and Selection Axes simultaneously (*Figure 5F*). Furthermore, on Maybe-Stop trials movement along the Selection axis overshot the level reached on No-Stop trials, as if overcompensating for the initial bias on this axis (*Figure 5A*, *Figure 5—figure supplement 2*). This overshoot was not seen for spontaneously-slower No-Stop trials (*Figure 5—figure supplement 2*). We conclude that variation in RT reflects multiple dynamic processes within basal ganglia circuits,

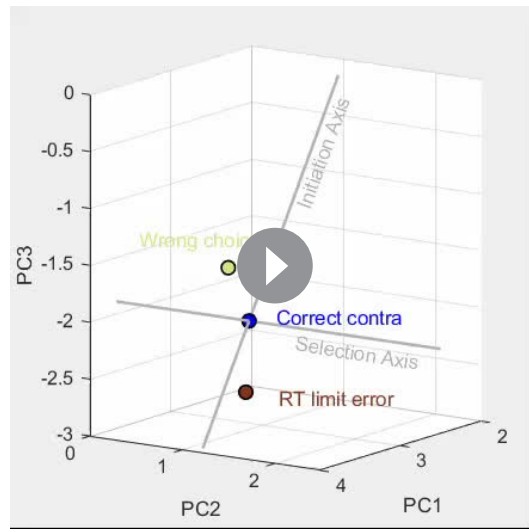

**Video 2.** State-space location at the Go cue varies with distinct types of errors.
https://elifesciences.org/articles/57215#video2

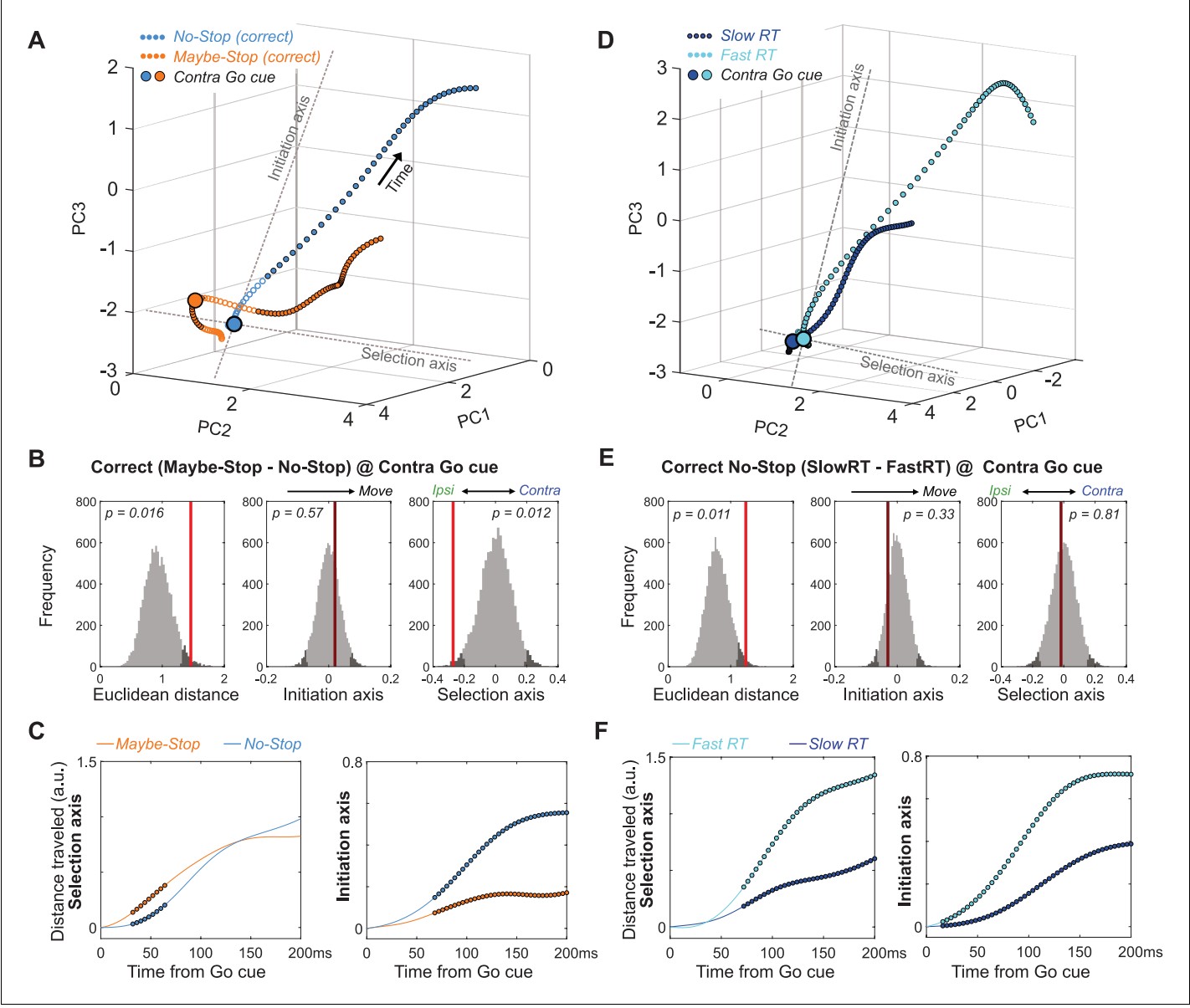

**Figure 5.** Multiple dynamics underlying slower reaction times. (**A**) Comparison of GP population state between correct Maybe-Stop (contra) and No-Stop (contra) trials (−100 to +250 ms around Go cue; same state-space and format as *Figures 3* and *4*). Time points of significant Euclidean separation between conditions are marked by filled circles. (**B**) Permutation tests (same format as *Figures 3* and *4*) comparing Maybe-Stop (contra) and No-Stop (contra) trials at the time of contra Go cue presentation. GP activity is significantly biased in the ipsi direction, when the contra-instructing cue might be followed by a Stop cue. (**C**) Examination of distance travelled after Go cue confirms that in the Maybe-Stop condition the trajectory first moves primarily along the Selection Axis (left), before making substantial progress along the Initiation Axis (right). (**D–F**) Same as A-C, but comparing correct contra No-Stop trials with faster or slower RTs (median split of RTs). Unlike Maybe-Stop trials, spontaneously slow RT trials do not show a starting bias (on either Initiation or Selection axes) and do not move on the Selection Axis before moving on the Initiation Axis.
The online version of this article includes the following figure supplement(s) for figure 5:

**Figure supplement 1.** Comparison of RT-matched Maybe-Stop and No-Stop trajectories.
**Figure supplement 2.** Comparison of Proactive and spontaneously Slow RT trajectories at movement onset.

with slowing due to proactive inhibition involving distinct internal control mechanisms to spontaneous RT variation.

Although reducing the dimensionality of data can be very useful for visualizing trajectories through state-space, we wished to ensure that our conclusions are not distorted by this procedure. We therefore repeated key analyses within the full 376-dimensional state space. Defining Initiation

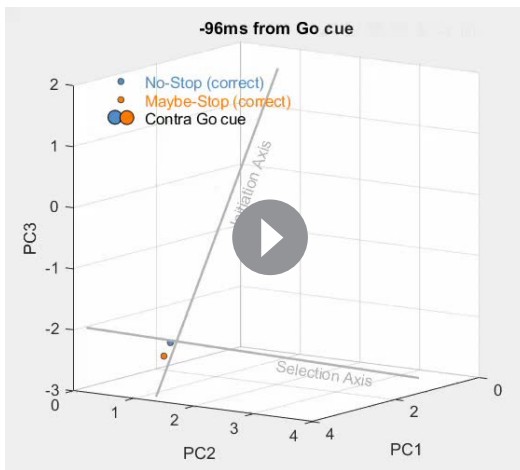

**Video 3.** Comparing neural trajectories for correct, contra Maybe-Stop versus No-Stop trials.
https://elifesciences.org/articles/57215#video3

and Selection Axes in the same way as before, but without the PCA step, produced essentially identical trajectory differences between conditions (*Figure 6*).

## Discussion

Stop-signal tasks are widely-used to test cognitive control (*Lipszyc and Schachar, 2010*), with proactive inhibition considered especially reliant on top-down, effortful, resource-demanding processes (*Jahanshahi et al., 2015*). Yet there have been extended debates about which psychological and neural mechanisms support proactive control (*Verbruggen and Logan, 2009*; *Chatham et al., 2012*; *Aron et al., 2014*; *Leunissen et al., 2016*). We have demonstrated here that a key behavioral signature of proactive control – selective slowing of RTs when a Stop signal is expected - can arise through multiple covert strategies. These are visible as changes to the dynamic state of GPe by the time of Go cue presentation, and include a bias towards an alternative action, and/or starting further from the 'point-of-no-return' in action initiation.

**Figure 6.** Defining Initiation, Selection Axes with or without prior dimension reduction. (**A**) Replotting major results from *Figures 3–5* in two dimensions. The Initiation and Selection Axes are defined as in the main figures, that is using points in the 10-D PC space. (**B**) Same as A, but defining axes in the full 376-D state space (skipping the PCA step).

Which internal strategies are employed for proactive inhibition is likely influenced by the specific experimental conditions (*Mayse et al., 2014*; *Yoshida et al., 2018*). For example, we used a brief limited hold period (800 ms) to encourage subjects to respond rapidly to the Go cue rather than waiting to see if the Stop cue is presented. This time pressure may have led rats to sometimes make guesses as to which cues will be presented, and position their neural state accordingly. We also used a task design with asymmetric (ipsi/contra) stop probabilities, to probe the selectivity of proactive inhibition (*Aron and Verbruggen, 2008*). Motivational aspects are known to be important in proactive inhibition (*Meyer and Bucci, 2016*): the ipsi bias we observed on the Selection axis on Maybe-Stop (contra) trials may partly reflect asymmetric reward expectancy (*Kawagoe et al., 1998*), simply because ipsi movements are more consistently rewarded from that state. Unlike human subjects, we cannot verbally instruct rats to perform the task in a certain way (although human cognitive strategies do not always follow experimenter intentions either). It might seem simpler, and less error-prone, for the rats to just select from the slower portion of their regular RT distribution. We suggest that they are unable to consistently do so, given the high spontaneous variability in RTs. The degree to which specific neural strategies are employed may also vary between rats; we found some preliminary evidence for this (*Figure 4—figure supplement 1*), though investigating this further would require more animals and more recorded cells in each animal.

The term 'proactive' or 'cognitive' control has been used to refer both to stop-signal tasks like this one, in which subjects are cued about the upcoming stop probability (e.g. *Cai et al., 2011*; *Jahfari et al., 2012*; *Zandbelt et al., 2013*), and also to uncued behavioral adjustments that subjects make after each trial (e.g. longer RTs following trials in which Stop cues occurred; *Chen et al., 2010*; *Pouget et al., 2011*; *Mayse et al., 2014*). Although not the focus of this study, our rats did slow down slightly on average after Stop trials or errors (*Figure 4—figure supplement 2A*). This slowing was associated with a modest shift on the Initiation Axis in the same, movement-opposed direction as in our main results (*Figure 4—figure supplement 2B*), but this effect did not reach significance. Thus both behavioral and neural data suggest that the cued component of proactive inhibition was substantially greater than post-trial adjustments under our particular task conditions.

Our ability to reveal distinct strategies for proactive inhibition relies on a dynamical systems approach with single-cell resolution. This method may be especially important for deciphering structures like GP, where projection neurons show continuous, diverse activity patterns. As intermingled GP neurons increased and decreased firing at each moment, the resulting network state changes would likely be undetected using aggregate measures such as photometry or fMRI. Speculatively, we suggest that an enhanced ability to make subtle adjustments to dynamical state may be part of the reason why GP projection neurons show high spontaneous activity, in contrast to (for example) the near-silence of most striatal projection neurons, most of the time.

Prior examinations of motor/premotor cortical dynamics during reaching movements in non-human primates have demonstrated distinct neural dimensions for movement preparation and execution ('*What*' to do) and movement triggering ('*When*' to do it) (*Elsayed et al., 2016*; *Kaufman et al., 2016*). Our Selection and Initiation axes are analogous, although our task lacks an explicit preparation epoch and has only two action choices (left vs. right). One notable difference in the non-human primate studies is that movement preparation occurred in distinct, orthogonal dimensions to movement execution, whereas we saw preparatory 'bias' along the same Selection axis that differentiated ipsi and contra trajectories during movement itself. Nonetheless, our observation that on correct Maybe-Stop trials, GP state evolved first along the Selection axis is consistent with evidence that movement preparation and movement initiation can be independent processes (*Haith et al., 2016*; *Thura and Cisek, 2017*), and that these can be differentially modulated by the basal ganglia and dopamine (*Leventhal et al., 2014*; *Manohar et al., 2015*). It also appears consistent with recent observations that, following an unexpected late change in target location, preparation dimensions are rapidly re-engaged (*Ames et al., 2019*).

The distinction between *What* and *When* dimensions is not readily compatible with sequential-sampling mathematical models of decision-making (*Smith and Ratcliff, 2004*; *Brown and Heathcote, 2008*; *Noorani and Carpenter, 2016*), which typically assume that RTs (*When*) directly reflect sufficient accumulation of evidence for a particular choice (*What*). Furthermore, when sensory cues are unambiguous the selection process appears to be much faster than standard RTs (*Stanford et al., 2010*; *Haith et al., 2016*). Why RTs are typically so much slower and more variable than required for sensory processing or action selection is not fully clear, but this extra time provides

opportunity for impulsive or inappropriate responses to be overruled, to increase behavioral flexibility.

The GPe is well positioned to contribute to such behavioral control. GPe has bidirectional connections with the subthalamic nucleus, a key component of the 'hyperdirect' pathway from frontal cortex that slows decision-making under conditions of conflict (*Cavanagh et al., 2011*). GPe itself is the target of the 'indirect' (striatopallidal) pathway, believed to discourage action initiation ('NoGo'; *Yoshida and Tanaka, 2016*; *Kravitz et al., 2010*), possibly due to pessimistic predictions of reward (*Collins and Frank, 2014*; *Kim et al., 2017*). In standard, firing rate-based models of basal ganglia function, GPe activity restrains actions by preventing pauses in the firing of basal ganglia output, that are in turn required to disinhibit movement-related activity in the brainstem and elsewhere (*Chevalier and Deniau, 1990*; *Roseberry et al., 2016*).

However, it is well-recognized that this model is too simple (*Gurney et al., 2001*; *Klaus et al., 2019*), and it does not account for the complex activity patterns within GPe that we and others have observed. For example, a straightforward application of the rate model might predict a systematic decrease in GPe firing rate with proactive inhibition, but we did not observe this (*Figure 2*), with the possible exception of trials with RT limit errors (*Figure 2—figure supplement 1*). Based on the current results, examining dimension-reduced population dynamics is a promising alternative approach for deciphering how subtle modulations in the firing of many basal ganglia neurons are coordinated to achieve behavioral functions.

At the same time, our study has several noteworthy limitations. Our reduction of complex dynamics to movement along Initiation and Selection axes is obviously a simplification. We did not record large populations of neurons simultaneously, which precludes effective analysis of neural dynamics on individual trials (*Afshar et al., 2011*). We did not classify GPe neurons by projection target (*Mallet et al., 2016*; *Abecassis et al., 2020*) largely because we did not consistently record sleep data to enable that classification (*Mallet et al., 2016*). We do not yet know the extent to which these population dynamics are shared with upstream (e.g. striatum) and downstream (e.g. substantia nigra pars reticulata) structures, which will be essential for elucidating how these dynamic changes actually influence behavior. Finally, we have not yet determined how the population dynamics reported here relate (if at all) to oscillatory dynamics reported in cortical-basal ganglia circuits during movement suppression (*Swann et al., 2009*; *Cavanagh et al., 2011*; *Leventhal et al., 2012*) and in pathological states such as Parkinson's Disease (*Hammond et al., 2007*). These are all worthy subjects for future investigation.

## Key resources

Rat (adult, male, Long-Evans, bred in-house).

## Materials and methods

All animal experiments were approved by the University of California, San Francisco Committee for the Use and Care of Animals. Adult male Long-Evans rats were housed on a 12 hr/12 hr reverse light-dark cycle, with training and testing performed during the dark phase.

### Behavior

Operant chambers (Med Associates, Fairfax VT) had five nose-poke holes on one wall, a food dispenser on the opposite wall, and a speaker located above the food port. The basic rat stop signal task has been previously described (*Leventhal et al., 2012.*; *Mallet et al., 2016*; *Schmidt et al., 2013*). At the start of each trial, one of the three more-central ports was illuminated ('Light On') indicating that the rat should poke in that port ('Center In') and wait. After a variable delay (500–1250 ms), a higher (4 kHz) or lower (1 kHz) pitch tone was presented for 50 ms ('Go Cue'), instructing a move to the adjacent port on the left or right side respectively. In Go trials (those without a Stop cue) if the rat left the initial center port ('Center Out') within 800 ms of Go cue onset, and then moved to the correct side port ('Side In') within 500 ms, a sugar pellet reward was delivered to the food dispenser with an audible click. As the rat left the center port, the center port light was turned off and both side port lights turned on. On Stop trials, the Go cue was followed by a Stop cue (white noise, 125 ms) with a short delay (the stop-signal delay, SSD). The SSD was randomly selected on each trial within a range (uniform distribution) of 100–200 ms (four rats) or 100–250 ms (two rats).

Stop trials were rewarded if the rat maintained its nose continuously within the start hole for a total of 800 ms after Go cue onset. Stop trials in which the rat initiated movement before the Stop cue began were converted into Go trials (i.e. no Stop cue was presented). Failed-Stop trials with RT >500 ms were excluded from electrophysiological analyses, since these were presumed to reflect trials for which rats successfully responded to the Stop cue, but then failed to maintain holding until reward delivery (see *Leventhal et al., 2012*; *Schmidt et al., 2013*; *Mayse et al., 2014*). Inter-trial intervals were randomly selected between 5 and 7 s. For included sessions, the median number of Go trials was 266 (range, 167–361) and the median number of Stop trials was 57 (range, 27–95).

To vary proactive inhibition, we changed the Stop cue probabilities between starting holes (as shown in *Figure 1*). The spatial mapping of probabilities was constant for each rat across sessions, but varied between rats. Within each session, the same start hole (and thus proactive condition) was repeated for 10–15 trials at a time. After ~3 months of training, rats showing consistent reaction time differences between Maybe-Stop and No-Stop conditions were eligible for electrode implantation.

## Electrophysiology

We report GP data from six rats (all animals in which we successfully recorded GP neurons during contraversive proactive inhibition). Each rat was implanted with 15 tetrodes (configured as independently-driveable bundles of 2–3 tetrodes, each within a polyimide tube with outer radius 140 μm), bilaterally targeting GP and substantia nigra reticulata (SNr). During task performance, wide-band (0.1–9000 Hz) electrophysiological data were recorded with a sampling rate of 30000/s using an Intan RHD2000 recording system (Intan Technologies). All signals were initially referenced to a skull screw (tip-flattened) on the midline 1 mm posterior to lambda. For spike detection we re-referenced to an electrode common average, and wavelet-filtered (*Wiltschko et al., 2008*) before thresholding. For spike sorting we performed automatic clustering units using MountainSort (*Chung et al., 2017*) followed by manual curation of clusters. Tetrodes were usually moved by 159 μm every 2–3 sessions. To avoid duplicate neurons we did not include data from the same tetrode across multiple sessions unless the tetrode had been moved by >100 μm between those sessions. Based on waveform and firing properties we further excluded an additional 25 units that appeared to be duplicates even though the tetrode had been moved. After recording was complete, we anesthetized rats and made small marker lesions by applying 10 μA current for 20 s for one or two wires of each tetrode. After perfusing the rats and slicing (at 40 μm) tissue sections were stained with cresyl violet and compared to the nearest atlas section (*Paxinos, 2006*).

## Data analysis

Smoothed firing rates were obtained by convolving each spike time with a Gaussian kernel (30 ms SD). Firing rates were normalized (Z-scored) using the neuron's session-wide mean and SD. Normalized average time series for contra and ipsi actions (500 ms each, around Center Out) were concatenated and used to construct a population activity matrix $\mathbf{R}$ = TC by N, with T = 251 (timepoints, at 2 ms intervals), C = 2 (ipsi/contra conditions), and N = 376 (the number of neurons). We subtracted the mean of each of the N columns to make data zero-centered, then performed principal components analysis (PCA) over matrix $\mathbf{R}$ using the MATLAB 'svd' function. Using the right singular vectors (*W*), we can calculate the PC scores (*S*) as $S = \mathbf{R}W$. For example, the first column of *S* contains the first principal component (PC1) over time, and the first column of *W* contains the weights for each of the N units for PC1. We used the first 10 PCs for analysis, and the Euclidean distance between conditions was compared in this 10-D space. The projections onto the Initiation or Selection Axes were calculated as the dot product of the state space position vector and the axis vector. State-space positions around the Go cue (or Stop cue) were calculated using the set of weights *W* to project the Go cue–aligned (or Stop cue-aligned) firing rates into the 10-D PC space. In other words, each neuron has a weight for each PC, and we calculate a net population position along each PC by multiplying each neuron's instantaneous firing rate by its weight, and summing across all neurons.

To test if whether state-space positions for two conditions (e.g. Successful- and Failed-Stops) are significantly separated, we ran permutation tests by randomly shuffling the trial conditions for each neuron (10000 shuffles for each test). Then, the distance in the population state space at each time

point was reconstructed using the firing rate differences between the shuffled trial averages for each condition. For example, if the mean FR of a unit (n) in surrogate Failed Stop trials (c1) and surrogate Successful Stop trials (c2) at Stop cue time (t) is $r_{(t,c1,n)}$ and $r_{(t,c2,n)}$, respectively, the difference between two conditions in k-dimension, $x_{(t,k)}$ is:

$$x_{(t,k)} = \sum_{n=1}^{N} \left( r_{(t,c1,n)} - r_{(t,c2,n)} \right) \times w_{(n,k)}$$

Repeated shuffling produces a surrogate data distribution for differences at each time point, and the original difference between conditions is compared to this distribution to determine statistical significance.

## Acknowledgements

We thank Michael Farries and Ali Mohebi for technical advice, Wei Wei and Vikaas Sohal for comments on the manuscript, and Alejandro Jimenez Rodriguez for discussions. This work was supported by NIH grants R01MH101697, R01NS078435 and R01DA045783, and the University of California, San Francisco.

## Additional information

### Funding

| Funder | Grant reference number | Author |
| --- | --- | --- |
| National Institute of Mental Health | R01 MH101697 | Joshua D Berke |
| National Institute on Drug Abuse | R01 DA045783 | Joshua D Berke |
| National Institute of Neurological Disorders and Stroke | R01NS078435 | Joshua D Berke |
| University of California, San Francisco | | Joshua D Berke |

The funders had no role in study design, data collection and interpretation, or the decision to submit the work for publication.

### Author contributions

Bon-Mi Gu, Conceptualization, Data curation, Software, Formal analysis, Investigation, Visualization, Methodology, Writing - review and editing; Robert Schmidt, Conceptualization, Visualization, Writing - review and editing; Joshua D Berke, Conceptualization, Resources, Supervision, Funding acquisition, Writing - original draft, Writing - review and editing

### Author ORCIDs

Joshua D Berke  https://orcid.org/0000-0003-1436-6823

### Ethics

Animal experimentation: All animal experiments were approved by the University of California, San Francisco Committee for the Use and Care of Animals (approval number: AN181071).

### Decision letter and Author response

Decision letter https://doi.org/10.7554/eLife.57215.sa1
Author response https://doi.org/10.7554/eLife.57215.sa2

## Additional files

### Supplementary files
• Transparent reporting form

### Data availability

The neurophysiology data and analysis code used in this study are available at figshare: https://doi.org/10.6084/m9.figshare.12367541.

The following dataset was generated:

| Author(s) | Year | Dataset title | Dataset URL | Database and Identifier |
|---|---|---|---|---|
| Gu BM, Schmidt R, Berke JD | 2020 | Globus pallidus dynamics reveal covert strategies for behavioral inhibition | https://doi.org/10.6084/m9.figshare.12367541 | figshare, 10.6084/m9.figshare.12367541 |

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
