## [Decision Letter]

**Acceptance summary:**

To study the role of the basal ganglia in response inhibition, the activity of neurons in the rat globus pallidus (GPe) was recorded while the animal performed a stop-signal task. The key analyses focused on the dynamics of population level activity, with the insight that different states could be identified that were associated with movement initiation and movements associated with a particular action. Most important, the state was context-dependent, shifting away from that associated with initiation and movement representation as the probability of response inhibition increased. In sum, the study applies sophisticated tools to reveal the dynamics of a key region of the brain involved in response inhibition.

**Decision letter after peer review:**

Thank you for submitting your article "Globus pallidus dynamics reveal covert strategies for behavioral inhibition" for consideration by *eLife*. Your article has been reviewed by three peer reviewers, including Daeyeol Lee as the Reviewing Editor and Reviewer #1, and the evaluation has been overseen by Richard Ivry as the Senior Editor.

The reviewers have discussed the reviews with one another and the Reviewing Editor has drafted this decision to help you prepare a revised submission.

As the editors have judged that your manuscript is of interest, but as described below that additional experiments are required before it is published, we would like to draw your attention to changes in our revision policy that we have made in response to COVID-19 (https://elifesciences.org/articles/57162). First, because many researchers have temporarily lost access to the labs, we will give authors as much time as they need to submit revised manuscripts. We are also offering, if you choose, to post the manuscript to bioRxiv (if it is not already there) along with this decision letter and a formal designation that the manuscript is “in revision at *eLife*”. Please let us know if you would like to pursue this option. (If your work is more suitable for medRxiv, you will need to post the preprint yourself, as the mechanisms for us to do so are still in development.)

Summary:

The authors have recorded the activity of neurons in the rat globus pallidus (GPe) while the animal performed a stop-signal task. The goal of this study was to examine the possible contribution of GPe in proactive inhibitory control whereby the animal could adjust the preparation of upcoming movement in order to reduce the initiation of undesired movements, namely, the movements that need to be canceled upon the arrival of stop signals. By examining the population activity in the state-space, the authors nicely demonstrated that increasing the probability of stop signal trials cause two different changes in GPe activity. First, the GPe population moved away from the region where movements can be triggered along the so-called initiation axis (along which the activity moves around the time of movement initiation). Second, the GPe activity moved away from the action that might be canceled along the different, so-called selection axis. The analyses and results are described clearly. Overall, the manuscript provides an important insight into the role of the basal ganglia in behavioral inhibition. Nevertheless, there are some interpretational limitations that the authors might need to discuss more.

Essential revisions:

1) One unfortunate consequence of the task design adopted in this study is that the position of the center port changes across 3 different conditions, and therefore could potentially influenced the activity of GPe neurons at the time of go cues. Can the authors exclude the possibility the changes in the GPe trajectory might result from the changes in the animal's body posture rather than from the preparatory bias for movement cancelation? Similarly, is it possible that activity during stop trial reflects the direction of the movement they need to make to get reward (i.e., the location of the food dispenser)? This would vary across trial-types (i.e., maybe- and no-stop trials) depending on the location of the central port.

2) The authors have used the phrase "cognitive control" (only) in the Abstract. While the results in this manuscript clearly provide important insights into the role of the basal ganglia in action selection and cancellation, however, it is unclear whether the type of control demonstrated in this study is analogous to cognitive control studied in human literature. In particular, there is no evidence that the adjustment is applied in response to the outcomes of recent trials, as in human and non-human primate studies. Another difference is that in this study, proactive control is applied directionally, which might not be the case in the standard stop signal task. How clearly can proactive "inhibitory" control be distinguished and separated from bias in action selection? These issues might need more careful discussion.

3) How will proactive control/state space approaches to these signals apply in instances of proactive control looking at trial history? For instance, as validation of the findings in this manuscript, can the authors show similar changes in neural firing/behavioral data on trials where rats experience multiple stops or goes in a row (e.g., conflict adaptation; post error slowing)? Presumably having just experienced a stop or an error, rats might slow their responding on the next trial and engage the strategies that the authors discuss. A similar story could be told for when there are multiple goes in a row.

4) Overall, some results can be explained well and more details can be included. Often results and analysis that explain fairly complicated figures are described in 1-2 sentences. Here are some examples.

a) It is not clear what Figure 2C,D is showing from the 1 sentence description in the text. It is said the contra and ipsi are equally represented but no statistic is given. It is not clear what "proactive control was associated with altered activity in different sets of GP neurons at different times" means and why the data is not shown.

b) It appears that the CDF in Figure 1D shows only the RT in non-stop (go) trials, whereas Figure 1B compares the RT in non-cancelled (error) stop trials, but this was not clearly explained. Also, it would be helpful to explain what RT and MT limit errors refer to at this point.

5) It is possible that GPe network state is not positioning anything strategically but is a reflection of upstream processing and/or the state that the body is in. Could GP activity states reflect that the animal has or is currently positioning its body in a certain way? Just because the rat is in the central port doesn't necessarily mean it is holding still or is not physically leaning in one direction or the other. My general feeling about these results is that the authors would get the exact same results in any task-related brain area using this analysis (e.g., dorsal striatum, SNr).

6) Figure 2 doesn't explain properly why these effects could not be observed at the level of single and population levels. It might be helpful to show firing rates for movements made in both directions for each of these figures. Also, how do these neurons and populations fire on errors (not moving quickly enough and going in the wrong direction). For the population histogram, the authors are averaging over all neurons, which may be problematic. From the heat plot it seems like half the neurons increase whereas the other half decrease firing during the trial. Averaging them all together would wash out any effects.

7) Some steps in the behavioral analysis might rest on assumptions that might not be valid. For example, it might be dangerous to group the stop trials where movements occur after stop cue together with go trials, since this strictly assumes that the go and stop processes are completely independent. The results from the primate literature (Schall) have shown that this assumption is not valid. In addition, it seems strange that stop trials with RT>500 ms were excluded. Were go trials with such long RT also removed from the analysis?

8) One interesting claim was that the trajectories of GP population activity were consistent with a spatial, not a temporal, offset along the initiation axis, but this was not entirely convincing. It appears that a temporal offset would require all (or most) of the neurons to have exactly the same delay. If only a subset of neurons showed this delay, or if the neurons showed different delays, wouldn't it also appear as a spatial offset? It might be useful to perform some simulations to document the sensitivity and specificity of using the PC-based analysis to uncover any relationship between neural activity and model predictions.

---

## [Author Response]

Essential revisions:1) One unfortunate consequence of the task design adopted in this study is that the position of the center port changes across 3 different conditions, and therefore could potentially influenced the activity of GPe neurons at the time of go cues. Can the authors exclude the possibility the changes in the GPe trajectory might result from the changes in the animal's body posture rather than from the preparatory bias for movement cancelation? Similarly, is it possible that activity during stop trial reflects the direction of the movement they need to make to get reward (i.e., the location of the food dispenser)? This would vary across trial-types (i.e., maybe- and no-stop trials) depending on the location of the central port.

We are confident that the variable spatial position of the start port is not driving our key results:

– We have previously demonstrated (Gage et al., 2010) that GPe neurons are very sensitive to movement direction, but are insensitive to start port spatial location. Those experiments used the same behavioral apparatus and similar task sequence (once again starting from any of the three center-most holes, as in this manuscript, but using Go trials only).

– We counterbalanced the spatial positions of the No-Stop, Maybe-Stop-Contra, and Maybe-Stop-Ipsi start ports between rats. The shifts we observed with contraversive proactive inhibition occurred in the same direction (i.e. a bias away from movement along the Initiation axis, and a bias away from contraversive action on the Selection axis), regardless of whether the Maybe-Stop-Contra port was physically located on the apparatus side congruent with the implant side, or on the opposite side.

2) The authors have used the phrase "cognitive control" (only) in the Abstract. While the results in this manuscript clearly provide important insights into the role of the basal ganglia in action selection and cancellation, however, it is unclear whether the type of control demonstrated in this study is analogous to cognitive control studied in human literature. In particular, there is no evidence that the adjustment is applied in response to the outcomes of recent trials, as in human and non-human primate studies. Another difference is that in this study, proactive control is applied directionally, which might not be the case in the standard stop signal task. How clearly can proactive "inhibitory" control be distinguished and separated from bias in action selection? These issues might need more careful discussion.

The terms “cognitive control” and “proactive inhibition” have been broadly applied in the human literature across a range of task designs. Our specific Maybe-Stop variant was directly inspired by, and based upon, human Maybe-Stop studies by Adam Aron and colleagues (e.g. Cai et al., 2011, Majid et al., 2013). Both those studies, and ours, were designed to study the movement-selectivity of proactive inhibition, using explicit cues in advance to prompt adjustments in stopping performance. Aron hypothesized (2011) that the GP was especially involved in movement-selective proactive inhibition, hence our choice to study GP here using a similar task design. We now make clear in the Abstract that we are studying selective proactive inhibition.

Our distinction between proactive effects on Initiation and Selection Axes is directly intended to help distinguish between inhibiting actions in general, and being biased in the preparation and selection of actions. We observed both effects. An interesting question is whether *any* situation that produces a directional bias is reflected in a similar GP population shift along the Selection Axis as defined here. We are currently investigating this using a probabilistically-rewarded task where contra and ipsi actions are associated with different reward expectation (see related part of Discussion).

It is true that “proactive inhibition” has also been used to refer to uncued behavioral adjustments after trial outcome (i.e. trial history-dependent effects). We observe this form of proactive inhibition as well, and it may involve overlapping neural mechanisms (see new Figure 4—figure supplement 2). We have now noted this in the text, and clarified the differences between these distinct manifestations of cognitive control in the Discussion.

3) How will proactive control/state space approaches to these signals apply in instances of proactive control looking at trial history? For instance, as validation of the findings in this manuscript, can the authors show similar changes in neural firing/behavioral data on trials where rats experience multiple stops or goes in a row (e.g., conflict adaptation; post error slowing)? Presumably having just experienced a stop or an error, rats might slow their responding on the next trial and engage the strategies that the authors discuss. A similar story could be told for when there are multiple goes in a row.

Our rats do indeed slow their responding slightly after stops or errors, and this is associated with a slight shift along the Initiation Axis in the same direction as in our main results (i.e. starting further from movement onset; Figure 4—figure supplement 2). However, the effect is smaller than the main proactive effects we report, and does not reach significance with our current data.

4) Overall, some results can be explained well and more details can be included. Often results and analysis that explain fairly complicated figures are described in 1-2 sentences. Here are some examples.a) It is not clear what Figure 2C,D is showing from the 1 sentence description in the text. It is said the contra and ipsi are equally represented but no statistic is given. It is not clear what "proactive control was associated with altered activity in different sets of GP neurons at different times" means and why the data is not shown.

It is true that we go relatively quickly over the individual neuron results in Figure 2; this is because the primary focus of the manuscript is the population dynamics and how they relate to behavior. Nonetheless, we have revised and extended the Results text to be hopefully more clear. We do give a statistical result for equal representation of contra and ipsi in Figure 2E (p=0.87) and have updated the legend to indicate that this is the same Wilcoxon signed rank test as in the previous panel.

The data that was previously not shown is now included in Figure 2—figure supplement 1C. This indicates that although some individual GP neurons significantly distinguish the Maybe-Stop and No-Stop conditions, we did not observe a neuronal subpopulation that maintains this distinction across extended periods (the timing of the Go cue is unpredictable). Our failure to observe such a subpopulation lead us to examine population dynamics.

b) It appears that the CDF in Figure 1D shows only the RT in non-stop (go) trials, whereas Figure 1B compares the RT in non-cancelled (error) stop trials, but this was not clearly explained. Also, it would be helpful to explain what RT and MT limit errors refer to at this point.

We have clarified Figure 1B to indicate that the RT distribution purple line refers specifically to Failed Stop trials (on Correct Stops, there is no RT to plot). We have noted in the legend that Figure 1D refers specifically to trials in which no Stop cue is presented, as the reviewer surmised. We have also added the definition of RT and MT limit errors to the Figure 1 legend.

5) It is possible that GPe network state is not positioning anything strategically but is a reflection of upstream processing and/or the state that the body is in. Could GP activity states reflect that the animal has or is currently positioning its body in a certain way? Just because the rat is in the central port doesn't necessarily mean it is holding still or is not physically leaning in one direction or the other. My general feeling about these results is that the authors would get the exact same results in any task-related brain area using this analysis (e.g., dorsal striatum, SNr).

Our task design exceeds prevailing standards in the rodent stop-signal field, as we constrain the rat to maintain head entry in the period preceding both go and stop cues. In any stop-signal study, regardless of species, one may be concerned that some unknown, uncontrolled aspect of behavior is contributing to the results. However, the specific character of our neural results (shifting away from contralateral movement initiation when contralateral movements may have to be cancelled) is very consistent with the proactive control interpretation that we favor.

Our GP results might indeed be mirrored in upstream and/or downstream structures, and determining whether this is the case is an important question we are addressing in ongoing research. Based on very preliminary data from SNr, it is not the case that the exact same results are present in any task-related brain area, but data collection for that study is not yet complete.

6) Figure 2 doesn't explain properly why these effects could not be observed at the level of single and population levels. It might be helpful to show firing rates for movements made in both directions for each of these figures. Also, how do these neurons and populations fire on errors (not moving quickly enough and going in the wrong direction). For the population histogram, the authors are averaging over all neurons, which may be problematic. From the heat plot it seems like half the neurons increase whereas the other half decrease firing during the trial. Averaging them all together would wash out any effects.

We address each of these points in turn in new Figure 2—figure supplement 1 panels:

– Panel B shows the firing rates for ipsi movements, in the same way we showed the contra movement in Figure 2C.

– Panel E shows the population average firing rates for Go-related errors: not moving quickly enough (RT limit error) and going in the wrong direction.

– Panel C shows the average activity separately for units with increases vs decreases in firing. We still do not see any significant difference between Maybe-Stop and No-Stop conditions as rats wait for the Go cue.

7) Some steps in the behavioral analysis might rest on assumptions that might not be valid. For example, it might be dangerous to group the stop trials where movements occur after stop cue together with go trials, since this strictly assumes that the go and stop processes are completely independent. The results from the primate literature (Schall) have shown that this assumption is not valid. In addition, it seems strange that stop trials with RT>500 ms were excluded. Were go trials with such long RT also removed from the analysis?

The behavioral analysis does not make this assumption. We now clarify (in Results and Figure 1D legend) that the RT comparison between Maybe-Stop and No-Stop trials includes only trials in which no Stop cue occurs.

The exclusion of Stop trials with RT>500ms serves to distinguish between “failures-to-stop” and “failures-to-wait”, in which the rat successfully cancels the action but then fails to sustain the hold for the required period (noted in Figure 1 legend). A prior, in-depth analysis (Mayse et al., 2014) demonstrated that it is important to distinguish between these, for example to correctly estimate stop-signal reaction times. In addition, we previously showed that this distinction between two types of errors is apparent in neural signatures of the stop process. For example, a pulse of beta oscillations is rapidly evoked by the Stop cue on Correct Stop trials, and Failed Stops with RT>500ms, but not on Failed Stops with RT<500ms (Leventhal et al., 2012). Finally, even we include RT>500ms stop failed trials, the key result on reactive stopping is unchanged (at the time of the Stop cue, GP position along the Initiation Axis is significantly different for Correct and Failed Stop trials; p<0.0001).

8) One interesting claim was that the trajectories of GP population activity were consistent with a spatial, not a temporal, offset along the initiation axis, but this was not entirely convincing. It appears that a temporal offset would require all (or most) of the neurons to have exactly the same delay. If only a subset of neurons showed this delay, or if the neurons showed different delays, wouldn't it also appear as a spatial offset? It might be useful to perform some simulations to document the sensitivity and specificity of using the PC-based analysis to uncover any relationship between neural activity and model predictions.

We now more clearly explain in the revised text that our key conceptual distinction is: does proactive inhibition involve an altered neural state *in advance* of the Go cue, or a change in neural information processing *subsequent* to Go cue onset? We provide compelling evidence for an altered neural state in advance of the Go cue, manifested as a spatial offset in state-space. Although interesting changes in information processing subsequent to Go cue onset can also occur (e.g. Figure 5—figure supplements 1 and 2) they cannot account for changes before the Go cue.